# Adversarial Robustness of Streaming Algorithms
# through Importance Sampling

**Vladimir Braverman** [1]  **Avinatan Hassidim** [1]  **Yossi Matias** [1]  **Mariano Schain** [1]  **Sandeep Silwal** [2]  **Samson Zhou** [3]

## Abstract

In the adversarial streaming model, an adversary gives an algorithm a sequence of adaptively chosen updates as a data stream and the goal of the algorithm is to compute or approximate some predetermined function for every prefix of the adversarial stream. However, the adversary may generate future updates based on previous outputs of the algorithm and in particular, the adversary may gradually learn the random bits internally used by an algorithm to manipulate dependencies in the input. This is especially problematic as many important problems in the streaming model require randomized algorithms, as they are known to not admit any deterministic algorithms that use sublinear space. In this paper, we introduce adversarially robust streaming algorithms for central machine learning and algorithmic tasks, such as regression and clustering, as well as their more general counterparts, subspace embedding, low-rank approximation, and coreset construction. Our results are based on a simple, but powerful, observation that many importance sampling-based algorithms give rise to adversarial robustness in contrast to sketching based algorithms, which are very prevalent in the streaming literature but suffer from adversarial attacks. In addition, we show that the well-known merge and reduce paradigm used for corset construction in streaming is adversarially robust. To the best of our knowledge, these are the first adversarially robust results for these problems yet require no new algorithmic implementations. Finally, we empirically confirm the robustness of our algorithms on various adversarial attacks and demonstrate that by contrast, some common existing algorithms are not robust.

## 1. Introduction

Robustness against adversarial attacks have recently been at the forefront of algorithmic design for machine learning tasks (Goodfellow et al., 2015; Carlini & Wagner, 2017; Athalye et al., 2018; Madry et al., 2018; Tsipras et al., 2019). We extend this line of work by studying adversarially robust streaming algorithms.

In the streaming model, data points are generated one at a time in a stream and the goal is to compute some meaningful function of the input points while using a limited amount of memory, typically *sublinear* in the total size of the input. The streaming model is applicable in many algorithmic and ML related tasks where the size of the data far exceeds the available storage. Applications of the streaming model include monitoring IP traffic flow, analyzing web search queries (Liu et al., 2016), processing large scientific data, feature selection in machine learning (Hou et al., 2021; Gomes et al., 2019; Wu et al., 2010), and estimating word statistics in natural language processing (Goyal et al., 2012) to name a few. Streaming algorithms have also been implemented in popular data processing libraries such as Apache Spark which have implementations for streaming tasks such as clustering and linear regression (Zaharia et al., 2016a).

In the adversarial streaming model (Mitrovic et al., 2017; Bogunovic et al., 2017; Avdiukhin et al., 2019; Ben-Eliezer & Yogev, 2020; Ben-Eliezer et al., 2020; Hassidim et al., 2020; Woodruff & Zhou, 2020; Alon et al., 2021; Kaplan et al., 2021), an adversary gives an algorithm a sequence of adaptively chosen updates $u_1, \ldots, u_n$ as a data stream. The goal of the algorithm is to compute or approximate some predetermined function for every prefix of the adversarial stream, but the adversary may generate future updates based on previous outputs of the algorithm. In particular, the adversary may gradually learn the random bits internally used by an algorithm to manipulate dependencies in the input. This is especially problematic as many important problems in the streaming model require randomized algorithms, as they are known to not admit any deterministic algorithms that use sublinear space. Studying when adversarially robust streaming algorithms are possible is an important problem in lieu of recent interest in adversarial attacks in ML with applications to adaptive data analysis.

Authors listed in alphabetical order. [1]Google [2]Electrical Engineering and Computer Science Department, Massachusetts Institute of Technology, Cambridge, MA, USA [3]Computer Science Department, Carnegie Mellon University, Pittsburgh, PA, USA. Correspondence to: Samson Zhou <samsonzhou@gmail.com>.

*Accepted by the ICML 2021 workshop on A Blessing in Disguise: The Prospects and Perils of Adversarial Machine Learning.* Copyright 2021 by the author(s).

**Related Works.** Adversarial robustness of streaming algorithms has been an important topic of recent research. On the positive note, (Ben-Eliezer et al., 2020) gave a robust framework for estimating the $L_p$ norm of points in a stream in the insertion-only model, where previous stream updates cannot later be deleted. Their work thus shows that deletions are integral to the attack of (Hardt & Woodruff, 2013). Subsequently, (Hassidim et al., 2020) introduced a new algorithmic design for robust $L_p$ norm estimation algorithms, by using differential privacy to protect the internal randomness of algorithms against the adversary. Although (Woodruff & Zhou, 2020) tightened these bounds, showing that essentially no losses related to the size of the input $n$ or the accuracy parameter $\varepsilon$ were needed, (Kaplan et al., 2021) showed that this may not be true in general. Specifically, they showed a separation between oblivious and adversarial streaming in the adaptive data analysis problem.

(Ben-Eliezer & Yogev, 2020) showed that sampling is not necessarily adversarially robust; they introduce an exponentially sized set system where a constant number of samples, corresponding to the VC-dimension of the set system, may result in a very unrepresentative set of samples. However, they show that with an additional logarithmic overhead in the number of samples, then Bernoulli and/or reservoir sampling are adversarially robust. This notion is further formalized by (Alon et al., 2021), who showed that the classes that are online learnable require essentially sample-complexity proportional to the Littlestone dimension of the underlying set system, rather than VC dimension. However, these sampling procedures are uniform in the sense that each item in the stream is sampled with the same probability. Thus the sampling probability of each item is *oblivious* to the identity of the item. By contrast, we show the robustness for a variety of algorithms based on *non-oblivious* sampling, where each stream item is sampled with probability roughly proportional to the "importance" of the item.

### 1.1. Our Contributions

Our main contribution is a powerful yet simple statement that algorithms based on non-oblivious sampling are adversarially robust if informally speaking, the process of sampling each item in the stream can be viewed as using fresh randomness independent of previous steps, even if the sampling probabilities depend on previous steps.

Let us describe, very informally, our meta-approach. Suppose we have an adversarial stream of elements given by $u_1, \ldots, u_n$. Our algorithm $\mathcal{A}$ will maintain a data structure $A_t$ at time $t$ which updates as the stream progresses. $\mathcal{A}$ will use a function $g(A_t, u_t)$ to determine the probability of sampling item $u_t$ to update $A_t$ to $A_{t+1}$. The function $g$ measures the "importance" of the element $u_t$ to the overall problem that we wish to solve. For example, if our applica-

tion is $k$-means clustering and $u_t$ is a point far away from all previously seen points so far, we want to sample it with a higher probability. We highlight that even though the sampling probability for $u_t$ given by $g(A_t, u_t)$ is adversarial, since the adversary designs $u_t$ and previous streaming elements, the *coin toss* performed by our algorithm $\mathcal{A}$ to keep item $u_t$ is *independent* of any events that have occurred so far, including the adversary's actions. This new randomness introduced by the independent coin toss is a key conceptual step in the analysis for all of the applications listed in Table 1.

Contrast this to the situation where a "fixed" data structure or sketch is specified upfront. In this case, we would not be adaptive to which inputs $u_t$ the adversary designs to be "important" for our problem which would lead us to potentially disregard such important items rendering the algorithm ineffective.

As applications of our meta-approach, we introduce adversarially robust streaming algorithms for two central machine learning tasks, regression and clustering, as well as their more general counterparts, subspace embedding, low-rank approximation, and coreset construction.

We show that several methods from the streaming algorithms "toolbox", namely merge and reduce, online leverage score sampling, and edge sampling are adversarially robust "for free." As a result, *existing* (and future) streaming algorithms that use these tools are robust as well. We discuss our results in more detail below and provide a summary of our results and applications in Table 1.

| Meta-approach | Applications |
|---|---|
| Merge and reduce (Theorem 1.1) | Coreset construction, SVMs, Gaussian mixture models, $k$-means/median clustering, projective clustering, PCA, $M$-estimators |
| Row sampling (Theorem 1.2) | Linear regression, spectral approximation, low-rank approximation, projection-cost preservation, $L_1$-subspace embedding |

*Table 1.* Summary of our robust sampling frameworks and corresponding applications

We first show that the well-known merge and reduce paradigm is adversarially robust. Since the merge and reduce paradigm defines coreset constructions, we thus obtain robust algorithms for $k$-means, $k$-median, Bregman clustering, projective clustering, principal component analysis (PCA), non-negative matrix factorization (NNMF) (Lucic & Krause, 2017).

**Theorem 1.1 (Merge and reduce is adversarially robust)**
*Given an offline $\varepsilon$-coreset construction, the merge and reduce framework gives an adversarially robust streaming construction for an $\varepsilon$-coreset with high probability.*

For regression and other numerical linear algebra related tasks, we consider the row arrival streaming model, in which the adversary generates a sequence of row vectors $\mathbf{a}_1, \ldots, \mathbf{a}_n$ in $d$-dimensional vector space. For $t \in [n]$, the $t$-th prefix of the stream induces a matrix $\mathbf{A}_t \in \mathbb{R}^{t \times d}$ with rows $\mathbf{a}_1, \ldots, \mathbf{a}_t$. We denote this matrix as $\mathbf{A}_t = \mathbf{a}_1 \circ \ldots \circ \mathbf{a}_t$ and define $\kappa$ to be an upper bound on the largest condition number[1] of the matrices $\mathbf{A}_1, \ldots, \mathbf{A}_n$.

**Theorem 1.2 (Row sampling is adversarially robust)**
*There is a row sampling based framework for adversarially robust streaming algorithms that at each $t \in [n]$:*

*(1) Outputs a matrix $\mathbf{M}_t$ such that $(1 - \varepsilon)\mathbf{A}_t^\top \mathbf{A}_t \preceq \mathbf{M}_t^\top \mathbf{M}_t \preceq (1 + \varepsilon)\mathbf{A}_t^\top \mathbf{A}_t$, while sampling $\mathcal{O}\left(\frac{d^2 \kappa}{\varepsilon^2} \log n \log \kappa\right)$ rows (spectral approximation/subspace embedding/linear regression/generalized regression).*

*(2) Outputs a matrix $\mathbf{M}_t$ such that for all rank $k$ orthogonal projection matrices $\mathbf{P} \in \mathbb{R}^{d \times d}, (1 - \varepsilon) \|\mathbf{A}_t - \mathbf{A}_t \mathbf{P}\|_F^2 \leq \|\mathbf{M}_t - \mathbf{M}_t \mathbf{P}\|_F^2 \leq (1 + \varepsilon) \|\mathbf{A}_t - \mathbf{A}_t \mathbf{P}\|_F^2$, while sampling $\mathcal{O}\left(\frac{dk\kappa}{\varepsilon^2} \log n \log^2 \kappa\right)$ rows (projection-cost preservation/low-rank approximation).*

*(3) Outputs a matrix $\mathbf{M}_t$ such that $(1 - \varepsilon) \|\mathbf{A}_t \mathbf{x}\|_1 \leq \|\mathbf{M}_t \mathbf{x}\|_1 \leq (1 + \varepsilon) \|\mathbf{A}_t \mathbf{x}\|_1$, while sampling $\mathcal{O}\left(\frac{d^2 \kappa}{\varepsilon^2} \log^2 n \log \kappa\right)$ rows ($L_1$ subspace embedding).*

Finally, we show that our analysis also applies to algorithms for graph sparsification in which edges are sampled according to their "importance". See the appendix for details.

**Sketching vs Sampling Algorithms.** A central tool for randomized streaming algorithms is the use of linear sketches. These methods maintain a data structure $f$ such that after the $(i+1)$-th input $x_i$, we can update $f$ by computing a linear function of $x_i$. Typically, these methods employ a random matrix. For example, if the input consists of vectors, sketching methods will use a random matrix to project the vector into a much smaller dimension space. In (Hardt & Woodruff, 2013), it was proved no linear sketch can approximate the $L_2$-norm within a polynomial multiplicative factor against such an adaptive adversary. In general, streaming algorithms that use sketching are highly susceptible to the type of attack described in (Hardt & Woodruff, 2013) where

---

[1]the ratio of the largest and smallest nonzero singular values

the adversary can effectively "learn" the kernel of the linear function used and send inputs along the kernel. For example, if an adversary knows the kernel of the random matrix used to project the input points, then by sending points that lie on the kernel of the matrix as inputs, the adversary can render the whole streaming algorithm useless.

One the other hand, we employ a different family of streaming algorithms that are based on *sampling* the input rather than *sketching* it. Surprisingly, this simple change allows one to automatically get many adversarially robust algorithms either "for free" or *without* new algorithmic overheads. For more information, see Section 1.1. We emphasize that while our techniques are not theoretically sophisticated, we believe its power lies in its simple message that **sampling is often superior to sketching for adversarial robustness**. In addition to downstream algorithmic and ML applications, this provides an interesting separation and trade-offs between the two paradigms; for non-adversarial inputs sketching often gives similar or better performance guarantees for many tasks (Bar-Yossef et al., 2001).

## 2. Experiments

To illustrate the robustness of importance sampling based streaming algorithms, we devise adversarial settings for clustering and linear regression. With respect to our adversarial setting, we show that the performance of a merge-and-reduce based streaming $k$-Means algorithm is robust while a popular streaming $k$-Means implementation (not based on importance sampling) is not. Similarly, we show the robustness superiority of a streaming linear regression algorithm based on row sampling over a popular streaming linear regression implementation and over sketching.

**Streaming $k$-means** In this adversarial clustering setting we consider a series of point batches where all points except those in the last batch are randomly sampled from a two dimensional standard normal distribution and points in the last batch similarly sampled but around a distant center (see the data points realization in both panels of Figure 1). We then feed the point sequence to `StreamingKMeans`, the streaming $k$-Means implementation of Spark (Zaharia et al., 2016b) the popular big-data processing framework. As illustrated in the left panel of Figure 1, the resulting two centers are both within the origin. Now, this result occurs regardless of the last batch's distance from the origin, implying that the per-sample loss performance of the algorithm can be made arbitrarily large. Alternatively, we used a merge and reduce based streaming $k$-Means algorithm and show that one of the resulting cluster centers is at the distant cluster (as illustrated in the right panel of Figure 1) thereby keeping the resulting per sample loss at the desired minimum. Specifically we use `Streamkm` an implementation of StreamKM++ (Ackermann et al., 2012) from the ClusOpt

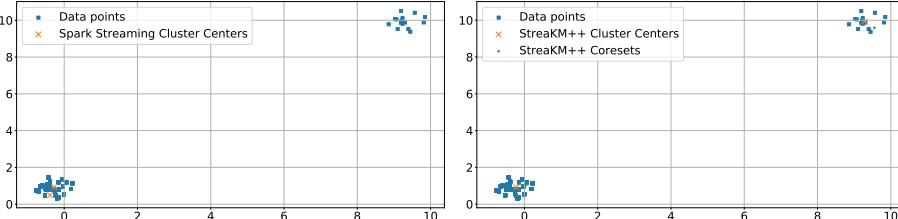

*Figure 1.* Cluster centers (x) on our adversarial datapoints setting for the popular Spark implementation (left) and StreamKM++ (right).

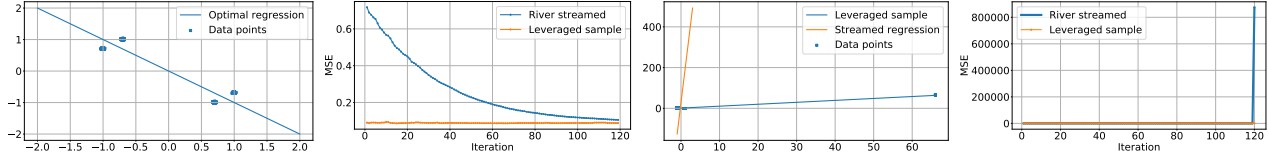

*Figure 2.* Streaming Linear Regression experiment (from left to right): points constellation without last batch, loss trajectory up to (not including) last batch, resulting regression lines upon training with last batch, and loss trajectory including the last batch.

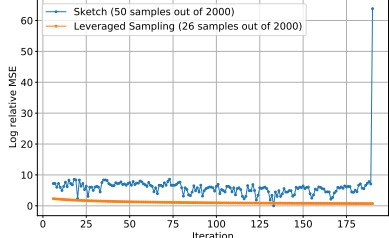

*Figure 3.* Comparing the performance trajectory of leveraged sampling Algorithm 1 to Sketching, for an adversarial data stream tailored to a sketching Matrix. Sketching performance deteriorates catastrophically upon the last batch, while leveraged sampling remains robust.

Core library (Macedo, 2020).

**Streaming linear regression.** Similar to the clustering setting, in the adversarial setting for streaming linear regression all batches except the last one are sampled around a constellation of four points in the plane such that the optimal regression line is of $-1$ slope through the origin (see the leftmost panel of Figure 2). The last batch of points however, is again far from the origin $(L, L)$ such that the resulting optimal regression line is of slope $1$ through the origin[2]. We compare the performance of `LinearRegression` from the popular streaming machine learning library River (Montiel et al., 2020) to our own row sampling based implementation of streaming linear regression along the lines of Algorithm 1 and observe the following: Without the last batch of points, both implementations result in the optimal regression line, however, the River implementation reaches that line only after several iterations, while our implementation is accurate throughout (This is illustrated in the second-left panel of Figure 2). When the last batch is

used, nevertheless, Algorithm 1 picks up the drastic change and adapts immediately to a line of the optimal slope (the blue line of the second right panel of Figure 2) while the River implementation update merely moves the line in the desired direction (the orange line in that same panel) but is far from catching up. Finally, the rightmost panel of Figure 2) details the loss trajectory for both implementations. While the River loss skyrockets upon the last batch, the loss of Algorithm 1 remains relatively unaffected, illustrating its adversarial robustness.

Note that in both the clustering and linear regression settings above, the adversary was not required to consider the algorithms internal randomization to achieve the desired effect (this is due to the local nature of the algorithms computations). This is not the case in the following setting.

**Sampling vs. sketching.** Finally, we compare the performance of the leverage sampling Algorithm 1 to Sketching. In this setting, for a random unit sketching matrix $S$ (that is, each of its elements is sampled from $\{-1, 1\}$ with equal probability), we create an adversarial data stream $A$ such that its columns are in the null space of $S$. As a result, the linear regression as applied to the sketched data $S \cdot A$ as a whole is unstable and might significantly differ from the resulting linear regression applied to streamed prefixes of the sketched data. As illustrated in Figure 3, this is not the case when applying the linear regression to the original streamed data $A$ using Algorithm 1. Upon the last batch, the performance of the sketching-based regression deteriorates by orders of magnitude, while the performance of Algorithm 1 is not affected. Moreover, the data reduction factor achieved by leveraged sampling[3] is almost double compared to the data reduction factor achieved by sketching.

---

[2]For MSE loss, this occurs for $L$ at least the square root of the number of batches.

[3]The original stream $A$ contained 2000 samples, each of dimension 10.

## Acknowledgments

Sandeep Silwal was supported in part by a NSF Graduate Research Fellowship Program. Samson Zhou was supported by a Simons Investigator Award of David P. Woodruff.

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

## A. Merge and Reduce

We show that the general merge and reduce paradigm is adversarially robust. Merge and reduce is widely used for the construction of a coreset, which provides dimensionality reduction on the size of an underlying dataset, so that algorithms for downstream applications can run more efficiently:

**Definition A.1** ($\varepsilon$ coreset) *Let $P \subset X$ be a set of elements from a universe $X$, $z \geq 0$, $\varepsilon \in (0,1)$, and $(P, \text{dist}, Q)$ be a query space. Then a subset $C$ equipped with a weight function $w : P \rightarrow \mathbb{R}$ is called an $\varepsilon$-coreset with respect to the query space $(P, \text{dist}, Q)$ if*

$$(1-\varepsilon) \sum_{\mathbf{p} \in P} \text{dist}(\mathbf{p}, Q)^z \leq \sum_{\mathbf{p} \in C} w(p) \text{dist}(\mathbf{p}, Q)^z$$
$$\leq (1+\varepsilon) \sum_{\mathbf{p} \in P} \text{dist}(\mathbf{p}, Q)^z.$$

The study of efficient offline coreset constructions for a variety of geometric and algebraic problems forms a long line of active research. For example, offline coreset constructions are known for linear regression, low-rank approximation, $L_1$-subspace embedding, $k$-means clustering, $k$-median clustering, $k$-center, support vector machine, Gaussian mixture models, $M$-estimators, Bregman clustering, projective clustering, principal component analysis, $k$-line center, $j$-subspace approximation, and so on. Thus, our result essentially shows that using the merge and reduce paradigm, these offline coreset constructions can be extended to obtain robust and accurate streaming algorithms. The merge and reduce paradigm works as follows. Suppose we have a stream $p_1, \ldots, p_n$ of length $n = 2^k$ for

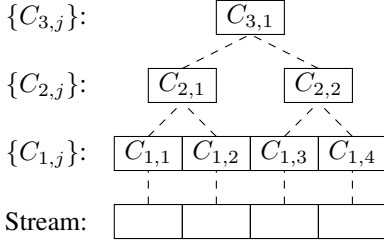

$\{C_{3,j}\}$:

$\{C_{2,j}\}$:

$\{C_{1,j}\}$:

Stream:

*Figure 4.* Merge and reduce framework. Each $C_{1,j}$ is an $\mathcal{O}\left(\varepsilon/\log n\right)$-coreset of the corresponding partition of the substream and each $C_{i,j}$ is an $\varepsilon$-coreset of $C_{i-1,2j-1}$ and $C_{i-1,2j}$ for $i > 1$.

some integer $k$, without loss of generality (otherwise we can use a standard padding argument to increase the length of the stream). Define $C_{0,j} = p_j$ for all $j \in [n]$. Consider $k$ levels, where each level $i \in [k]$ consists of $\frac{n}{2^i}$ coresets $C_{i,1}, \ldots, C_{i,n/2^i}$ and each coreset $C_{i,j}$ is an $\frac{\varepsilon}{2k}$-coreset of $C_{i-1,2j-1}$ and $C_{i-1,2j}$. Note that this approach can be implemented efficiently in the streaming model, since each $C_{i,j}$ can be built immediately once $C_{i-1,2j-1}$ and $C_{i-1,2j}$ are constructed, and after $C_{i,j}$ is constructed, then both $C_{i-1,2j-1}$ and $C_{i-1,2j}$ can be discarded. For an illustration of the merge and reduce framework, see Figure 4, though we defer all formal proofs to Section D. Using the coresets of (Braverman et al., 2020), Theorem 1.1 gives the following applications:

**Theorem A.2** *There exists a merge-and-reduce row sampling based framework for adversarially robust streaming algorithms that at each time $t \in [n]$:*

*(1) Outputs a matrix $\mathbf{M}_t$ such that $(1 - \varepsilon)\mathbf{A}_t^\top \mathbf{A}_t \preceq \mathbf{M}_t^\top \mathbf{M}_t \preceq (1 + \varepsilon)\mathbf{A}_t^\top \mathbf{A}_t$, while sampling $\mathcal{O}\left(\frac{d^2}{\varepsilon^2}\log^4 n \log \kappa\right)$ rows (spectral approximation/subspace embedding/linear regression/generalized regression).*

*(2) Outputs a matrix $\mathbf{M}_t$ such that for all rank $k$ orthogonal projection matrices $\mathbf{P} \in \mathbb{R}^{d \times d}$,*

$$(1 - \varepsilon)\|\mathbf{A}_t - \mathbf{A}_t\mathbf{P}\|_F^2 \le \|\mathbf{M}_t - \mathbf{M}_t\mathbf{P}\|_F^2$$
$$\le (1 + \varepsilon)\|\mathbf{A}_t - \mathbf{A}_t\mathbf{P}\|_F^2,$$

*while sampling $\mathcal{O}\left(\frac{k}{\varepsilon^2}\log^4 n \log^2 \kappa\right)$ rows (projection-cost preservation/low-rank approximation).*

*(3) Outputs a matrix $\mathbf{M}_t$ such that $(1 - \varepsilon)\|\mathbf{A}_t\mathbf{x}\|_1 \le \|\mathbf{M}_t\mathbf{x}\|_1 \le (1 + \varepsilon)\|\mathbf{A}_t\mathbf{x}\|_1$, while sampling $\mathcal{O}\left(\frac{d}{\varepsilon^2}\log^5 n \log \kappa\right)$ rows ($L_1$ subspace embedding).*

Using coresets of (Huang & Vishnoi, 2020), then Theorem 1.1 also gives applications for $(k,z)$-clustering such as $k$-median for $z = 1$ and $k$-means for $z = 2$. Moreover, (Lucic & Krause, 2017) noted that constructions of (Feldman & Langberg, 2011) give coresets for Bregman

clustering, which handles $\mu$-similar Bregman divergences such as the Itakura-Saito distance, KL-divergence, Mahalanobis distance, etc.

**Theorem A.3** *There exists a merge-and-reduce importance sampling based framework for adversarially robust streaming algorithms that at each time $t$:*

*(1) Outputs a set of centers that gives a $(1 + \varepsilon)$-approximation to the optimal $(k, z)$-clustering, $k$-means clustering $(z = 2)$, and $k$-median clustering $(z = 1)$, while storing $\mathcal{O}\left(\frac{1}{\varepsilon^{2z+2}}k\log^{2z+2}n\log k \log \frac{k \log n}{\varepsilon}\right)$ points.*

*(2) Outputs a set of centers that gives a $(1 + \varepsilon)$-approximation to the optimal $k$-Bregman clustering, while storing $\mathcal{O}\left(\frac{1}{\varepsilon^2}dk^3\log^3 n\right)$ points.*

Using the sensitivity bounds of (Varadarajan & Xiao, 2012a;b) and the coreset constructions of (Braverman et al., 2016), then Theorem 1.1 also gives applications for the following shape fitting problems:

**Theorem A.4** *There exists a merge-and-reduce importance sampling based framework for adversarially robust streaming algorithms that at each time $t$:*

*(1) Outputs a set of lines that gives a $(1+\varepsilon)$-approximation to the optimal $k$-lines clustering, while storing $\mathcal{O}\left(\frac{d}{\varepsilon^2}f(d,k)k^{f(d,k)}\log^4 n\right)$ points of $\mathbb{R}^d$, for a fixed function $f(d, k)$.*

*(2) Outputs a subspace that gives a $(1+\varepsilon)$-approximation to the optimal dimension $j$ subspace approximation, while storing $\mathcal{O}\left(\frac{d}{\varepsilon^2}g(d,j)k^{g(d,j)}\log^4 n\right)$ points of $\mathbb{R}^d$, for a fixed function $g(d, j)$.*

*(3) Outputs a set of subspaces that gives a $(1 + \varepsilon)$-approximation to the optimal $(j, k)$-projective clustering, while storing $\mathcal{O}\left(\frac{d}{\varepsilon^2}h(d,j,k)\log^3 n(\log n)^{h(d,j,k)}\right)$ points of $\mathbb{R}^d s$, for a fixed function $h(d, j, k)$, for a set of input points with integer coordinates.*

Adversarially robust approximation algorithms for Bayesian logistic regression, Gaussian mixture models, generative adversarial networks (GANs), and support vector machine can be obtained from Theorem 1.1 and coreset constructions of (Huggins et al., 2016; Feldman et al., 2019; Sinha et al., 2020; Tukan et al., 2020); a significant number of additional applications of Theorem 1.1 using coreset constructions can be seen from recent works and surveys on coresets, e.g., see (Lucic & Krause, 2017; Feldman, 2020).

## B. Adversarial Robustness of Subspace Embedding and Applications

We use $[n]$ to represent the set $\{1, \ldots, n\}$ for an integer $n > 0$. We typically use bold font to denote vectors and matrices. For a matrix $\mathbf{A}$, we use $\mathbf{A}^{-1}$ to denote the Moore-Penrose inverse of $\mathbf{A}$. We first formally define the goals of our algorithms:

**Problem B.1 (Spectral Approximation)** *Given a matrix* $\mathbf{A} \in \mathbb{R}^{n \times d}$ *and an approximation parameter* $\varepsilon > 0$, *the goal is to output a matrix* $\mathbf{M} \in \mathbb{R}^{m \times d}$ *with* $m \ll n$ *such that* $(1-\varepsilon) \|\mathbf{A}\mathbf{x}\|_2 \leq \|\mathbf{M}\mathbf{x}\|_2 \leq (1+\varepsilon) \|\mathbf{A}\mathbf{x}\|_2$ *for all* $\mathbf{x} \in \mathbb{R}^d$ *or equivalently,* $(1-\varepsilon) \mathbf{A}^\top \mathbf{A} \preceq \mathbf{M}^\top \mathbf{M} \preceq (1+\varepsilon) \mathbf{A}^\top \mathbf{A}$.

We note that linear regression is a well-known specific application of spectral approximation.

**Problem B.2 (Projection-Cost Preservation)** *Given a matrix* $\mathbf{A} \in \mathbb{R}^{n \times d}$, *a rank parameter* $k > 0$, *and an approximation parameter* $\varepsilon > 0$, *the goal is to find a matrix* $\mathbf{M} \in \mathbb{R}^{m \times d}$ *with* $m \ll n$ *such that for all rank* $k$ *orthogonal projection matrices* $\mathbf{P} \in \mathbb{R}^{d \times d}$,

$$(1-\varepsilon) \|\mathbf{A} - \mathbf{A}\mathbf{P}\|_F^2 \leq \|\mathbf{M} - \mathbf{M}\mathbf{P}\|_F^2 \leq (1+\varepsilon) \|\mathbf{A} - \mathbf{A}\mathbf{P}\|_F^2.$$

Note if $\mathbf{M}$ is a projection-cost preservation of $\mathbf{A}$, then its best low-rank approximation can be used to find a projection matrix that gives an approximation of the best low-rank approximation to $\mathbf{A}$.

**Problem B.3 (Low-Rank Approximation)** *Given a matrix* $\mathbf{A} \in \mathbb{R}^{n \times d}$, *a rank parameter* $k > 0$, *and an approximation parameter* $\varepsilon > 0$, *find a rank* $k$ *matrix* $\mathbf{M} \in \mathbb{R}^{n \times d}$ *such that* $(1 - \varepsilon) \left\|\mathbf{A} - \mathbf{A}_{(k)}\right\|_F^2 \leq \|\mathbf{A} - \mathbf{M}\|_F^2 \leq (1 + \varepsilon) \left\|\mathbf{A} - \mathbf{A}_{(k)}\right\|_F^2$, *where* $\mathbf{A}_{(k)}$ *for a matrix* $\mathbf{A}$ *denotes the best rank* $k$ *approximation to* $\mathbf{A}$.

**Problem B.4 ($L_1$-Subspace Embedding)** *Given a matrix* $\mathbf{A} \in \mathbb{R}^{n \times d}$ *and an approximation parameter* $\varepsilon > 0$, *the goal is to output a matrix* $\mathbf{M} \in \mathbb{R}^{m \times d}$ *with* $m \ll n$ *such that* $(1 - \varepsilon) \|\mathbf{A}\mathbf{x}\|_1 \leq \|\mathbf{M}\mathbf{x}\|_1 \leq (1 + \varepsilon) \|\mathbf{A}\mathbf{x}\|_1$ *for all* $\mathbf{x} \in \mathbb{R}^d$.

We consider the general class of row sampling algorithms, e.g., (Cohen et al., 2016; Braverman et al., 2020). Here we maintain a $L_p$ subspace embedding of the underlying matrix by approximating the online $L_p$ sensitivities of each row as a measure of importance to perform sampling. For more details, see Algorithm 1.

**Definition B.5 (Online $L_p$ Sensitivities)** *For a matrix* $\mathbf{A} = \mathbf{a}_1 \circ \ldots \circ \mathbf{a}_n \in \mathbb{R}^{n \times d}$, *the online sensitivity of row* $\mathbf{a}_i$ *for each* $i \in [n]$ *is the quantity* $\max_{\mathbf{x} \in \mathbb{R}^d} \frac{|\langle \mathbf{a}_i, \mathbf{x} \rangle|^p}{\|A_i \mathbf{x}\|_p^p}$, *where* $\mathbf{A}_{i-1} = \mathbf{a}_1 \circ \ldots \circ \mathbf{a}_{i-1}$.

---

**Algorithm 1** Row sampling based algorithms, e.g., (Cohen et al., 2016; Braverman et al., 2020)

---

**Input:** A stream of rows $\mathbf{a}_1, \ldots, \mathbf{a}_n \in \mathbb{R}^d$, parameter $p > 0$, and an accuracy parameter $\varepsilon > 0$
**Output:** A $(1 + \varepsilon)$ $L_p$ subspace embedding.
1: $\mathbf{M} \leftarrow \emptyset$
2: $\alpha \leftarrow \frac{Cd}{\varepsilon^2} \log n$ with sufficiently large parameter $C > 0$
3: **for** each row $\mathbf{a}_i$, $i \in [n]$ **do**
4:     **if** $\mathbf{a}_i \in \text{span}(\mathbf{M})$ **then**
5:         $\tau_i \leftarrow 2 \cdot \max_{\mathbf{x} \in \mathbb{R}^d, \mathbf{x} \in \text{span}(\mathbf{M})} \frac{|\langle \mathbf{a}_i, \mathbf{x} \rangle|^p}{\|\mathbf{M}\mathbf{x}\|_p^p + |\langle \mathbf{a}_i, \mathbf{x} \rangle|^p}$
    ▷See Remark B.6
6:     **else**
7:         $\tau_i \leftarrow 1$
8:     $p_i \leftarrow \min(1, \alpha\tau_i)$
9:     With probability $p_i$, $\mathbf{M} \leftarrow \mathbf{M} \circ \frac{\mathbf{a}_i}{p_i^{1/p}}$    ▷Online sensitivity sampling
10: **return** $\mathbf{M}$

---

We remark on standard computation or approximation of the online $L_p$ sensitivities, e.g., see (Cohen et al., 2015; 2016; 2017; Braverman et al., 2020).

**Remark B.6** *We note that for* $p = 1$, *a constant fraction approximation to any online* $L_p$ *sensitivity* $\tau_i$ *such that* $\tau_i > \frac{1}{\text{poly}(n)}$ *can be computed in polynomial time using (offline) linear programming while for* $p = 2$, $\tau_i$ *is equivalent to the online leverage score of* $\mathbf{a}_i$, *which has the closed form expression* $\mathbf{a}_i^\top (\mathbf{A}_i^\top \mathbf{A}_i)^{-1} \mathbf{a}_i$, *which can be approximated by* $\mathbf{a}_i^\top (\mathbf{M}^\top \mathbf{M})^{-1} \mathbf{a}_i$, *conditioned on* $\mathbf{M}$ *being a good approximation to* $\mathbf{A}_{i-1}$ *when* $\mathbf{a}_i$ *is in the span of* $\mathbf{M}$. *Otherwise,* $\tau_i$ *takes value 1 when* $\mathbf{a}_i$ *is not in the span of* $\mathbf{M}$.

**Lemma B.7 (Adversarially Robust $L_p$ Subspace Embedding and Linear Regression)** *Given* $\varepsilon > 0$, $p \in \{1, 2\}$, *and a matrix* $\mathbf{A} \in \mathbb{R}^{n \times d}$ *whose rows* $\mathbf{a}_1, \ldots, \mathbf{a}_n$ *arrive sequentially in a stream with condition number at most* $\kappa$, *there exists an adversarially robust streaming algorithm that outputs a* $(1 + \varepsilon)$ *spectral approximation with high probability. The algorithm samples* $\mathcal{O}\left(\frac{d^2 \kappa^2}{\varepsilon^2} \log n \log \kappa\right)$ *rows for* $p = 2$ *and* $\mathcal{O}\left(\frac{d^2 \lambda^2}{\varepsilon^2} \log^2 n \log \kappa\right)$ *rows for* $p = 1$, *with high probability, where* $\lambda$ *is a ratio between upper and lower bounds on* $\|\mathbf{A}\|_1$.

We also show robustness of row sampling for low-rank approximation by using online ridge-leverage scores. Together, Lemma B.7 and Lemma B.8 give Theorem 1.2.

**Lemma B.8** *(Adversarially Robust Low-Rank Approximation)*

*Given accuracy parameter $\varepsilon > 0$, rank parameter $k > 0$, and a matrix $\mathbf{A} \in \mathbb{R}^{n \times d}$ whose rows $\mathbf{a}_1, \ldots, \mathbf{a}_n$ arrive sequentially in a stream with condition number at most $\kappa$, there exists an adversarially robust streaming algorithm that outputs a $(1 + \varepsilon)$ low-rank approximation with high probability. The algorithm samples $\mathcal{O}\left(\frac{kd\kappa^2}{\varepsilon^2} \log n \log \kappa\right)$ rows with high probability.*

## C. Graph Sparsification

In this section, we highlight how the sampling paradigm gives rise to an adversarially robust streaming algorithm for graph sparsification. First, we motivate the problem of graph sparsification. Massive graphs arise in many theoretical and applied settings, such as in the analysis of large social or biological networks. A key bottleneck in such analysis is the large computational resources, in both memory and time, needed. Therefore, it is desirable to get a representation of graphs that take up far less space while still preserving the underlying "structure" of the graph. Usually the number of vertices is much fewer than the number of edges; for example in typical real world graphs, the number of vertices can be several orders of magnitude smaller than the number of edges (for example, see the graph datasets in (Rossi & Ahmed, 2015)). Hence, a natural benchmark is to reduce the number of edges to be comparable to the number of vertices.

The most common notion of graph sparsification is that of preserving the value of *all* cuts in the graph by keeping a small weighted set of edges of the original graph. More specifically, suppose our graph is $G = (V, E)$ and for simplicity assume all the edges have weight 1. A cut of the graph is a partition of $V = (C, V \setminus C)$ and the value of a cut, $\text{Val}_G(C)$, is defined as the number of edges that cross between the vertices in $C$ and $V \setminus C$. A graph $H$ on the same set of vertices as $V$ is a sparsifier if it preserves the value of every cut in $G$ and has a few number of weighted edges. For a precise formulation, see Problem C.1.

In addition to being algorithmically tractable, this formulation is natural since it preserves the underlying cluster structure of the graph. For example, if there are two well connected components separated by a sparse cut, i.e. two distinct communities, then the sparsifier according to the definition above will ensure that the two communities are still well separated. Conversely, by considering any cut within a well connected component, it will also ensure that any

community remains well connected (for more details, see (Satuluri et al., 2011) and references therein). Lastly, graph sparsification has been considered in other frameworks such as differential privacy (Eliás et al., 2020), distributed optimization (Wangni et al., 2018), and even learning graph sparsification using deep learning methods (Zheng et al., 2020). The formal problem definition of graph sparsification is as follows.

**Problem C.1 (Graph Sparsification)** *Given a graph weighted $G = (V, E)$ with $|V| = n, |E| = m$, and an approximation parameter $\varepsilon > 0$, compute a weighted subgraph $H$ of $G$ on the same set of vertices such that*

*(1) every cut in $H$ has value between $1 - \varepsilon$ and $1 + \varepsilon$ times its value in $G$: $(1 - \varepsilon)\text{Val}_G(C) \leq \text{Val}_H(C) \leq (1 + \varepsilon)\text{Val}_G(C)$ for all cuts $C$ where $\text{Val}_G(C), \text{Val}_H(C)$ denotes the cost of the cut in the graphs $G$ and $H$ respectively and for the latter quantity, the edges are weighted,*

*(2) the number of edges in $H$ is $\mathcal{O}\left(\frac{n \log n}{\varepsilon^2}\right)$.*

Ignoring dependence on $\varepsilon$, there are previous results that already get sparsifiers $H$ with $O(n \log n)$ edges (Benczúr & Karger, 1996; Spielman & Srivastava, 2008). Their setting is when the *entire* graph is present up-front in memory. In contrast, we are interested in the streaming setting where future edges can depend on past edges as well as revealed randomness of an algorithm while processing the edges.

Our main goal is to show that the streaming algorithm from (Ahn & Guha, 2009) (presented in Algorithm 2 in Section F), which uses a sampling procedure to sample edges in a stream, is adversarially robust, albeit with a slightly worse guarantee for the number of edges. Following the proof techniques of the non streaming algorithm given in (Benczúr & Karger, 1996), it is shown in (Ahn & Guha, 2009) that Algorithm 2 outputs a subgraph $H$ such that $H$ satisfies the conditions of Problem C.1 with probability $1 - 1/\text{poly}(n)$ where the probability can be boosted by taking a larger constant $C$. We must show that this still holds true if the edges of the stream are adversarially chosen, i.e., when **new edges in the stream depend on the previous edges and the randomness used by the algorithm so far.** We thus again use a martingale argument; the full details are given in Section F. As in Section B, we let $\kappa_1$ and $\kappa_2$ to be deterministic lower/upper bounds on the size of any cut in $G$ and define $\kappa = \kappa_2/\kappa_1$.

**Theorem C.2** *Given a weighted graph $G = (V, E)$ with $|V| = n$ whose edges $e_1, \ldots, e_m$ arrive sequentially in a stream, there exists an adversarially robust streaming algorithm that outputs a $1 \pm \varepsilon$ cut sparsifier with $\mathcal{O}\left(\frac{\kappa^2 n \log n}{\varepsilon^2}\right)$ edges with probability $1 - 1/\text{poly}(n)$.*

## D. Missing Proofs from Section A

In this section, we give the full details of the statements in Section A. Coreset constructions are known for a variety of problems, e.g., in computational geometry (Feldman et al., 2010; Feldman & Langberg, 2011; Braverman et al., 2016; Lucic & Krause, 2017; Sohler & Woodruff, 2018; Braverman et al., 2019; Huang & Vishnoi, 2020; Feldman, 2020), linear algebra (Braverman et al., 2020), machine learning (Munteanu et al., 2018; Baykal et al., 2019; Mussay et al., 2020). We first show that coreset construction is adversarially robust by considering the merge and reduce framework. For example, consider the offline coreset construction through sensitivity sampling.

**Lemma D.1 (Lemma 2.3 in (Lucic & Krause, 2017))**
*Given $\varepsilon > 0$ and $\delta \in (0,1)$, let $P$ be a set of weighted points, with non-negative weight function $\mu : P \to \mathbb{R}_{\geq 0}$ and let $s : P \to \mathbb{R}^{\geq 0}$ denote an upper bound on the sensitivity of each point. For $S = \sum_{p \in P} \mu(p)s(p)$, let $m = \Omega\left(\frac{S^2}{\varepsilon^2}\left(d' + \log\frac{1}{\delta}\right)\right)$, where $d'$ is the pseudo-dimension of the query space. Let $C$ be a sample of $m$ points from $P$ with replacement, where each point $p \in P$ is sampled with probability $q(p) = \frac{\mu(p)s(p)}{S}$ and assigned the weight $\frac{\mu(p)}{m \cdot q(p)}$ if sampled. Then $C$ is an $\varepsilon$-coreset of $P$ with probability at least $1 - \delta$.*

We first observe that any streaming algorithm that uses linear memory is adversarially robust because intuitively, it can recompute an exact or approximate solution at each step.

**Lemma D.2** *Given a set of points $P$, there exists an offline adversarially robust construction that outputs an $\varepsilon$-coreset of $P$ with probability at least $1 - \delta$.*

**Proof :** Given an adversary $A$, let $P = p_1, \ldots, p_n$ be a set of points such that each $p_i$ with $i \in [n]$ is generated by $A$, possibly as a function of $p_1, \ldots, p_{i-1}$. For example, it may be possible that the points $p_1, \ldots, p_{n/2}$ are a coreset of some set of points $P_1$ and the points $p_{n/2+1}, \ldots, p_n$ (1) were either generated with full knowledge of $p_1, \ldots, p_{n/2}$ or (2) are a coreset of a set of points $P_2$ generated with full knowledge of $P_1$. Let $s(p)$ be an upper bound on the sensitivity of each point in $P$ and consider the sensitivity sampling procedure described in Lemma D.1. We would like to sample each point with probability $q(p)$. Each point in $C$ is chosen to be $p$ with probability $q(p)$. However, if our algorithm generates internal randomness to perform this sampling procedure, it may be possible for an adversary to either learn correlations with the internal randomness or even learn the internal randomness entirely (such as the seed of a pseudorandom generator). Thus the choice for each point of $C$ may no longer be independent, so we are no longer guaranteed that the resulting construction is a coreset.

Instead, suppose the randomness used by the algorithm at time $i$ in the sampling procedure is independent of the choices of $p_1, \ldots, p_{i-1}$, e.g., the algorithm has access to a source of fresh public randomness at each time in the data stream. Then the algorithm can generate $C$ independent of the choices of $p_1, \ldots, p_{i-1}$. Thus by Lemma D.1, $C$ is an $\varepsilon$-coreset of $P$ with probability at least $1 - \delta$. □

We emphasize that Lemma D.2 shows that any offline coreset construction is adversarially robust; the example of sensitivity sampling is specifically catered to our applications of the merge and reduce framework to clustering.

We now prove our main statement.

**Proof of Theorem 1.1:** Let $\delta = \frac{1}{\text{poly}(n)}$ and consider an $\varepsilon$-coreset construction with failure probability $\delta$. We prove that the merge and reduce framework gives an adversarially robust construction for an $\varepsilon$-coreset with probability at least $1 - 2n\delta$. We consider a proof by induction on an input set $P$ of $n$ points, supposing that $n = 2^k$ for some integer $k > 0$. Observe that $C_{0,j}$ is a coreset of $p_j$ for $j \in [n]$ since $C_{0,j} = p_j$. Let $\mathcal{E}_i$ be the event that for a fixed $i \in [k]$ that $C_{i-1,j}$ is an $\frac{\varepsilon}{2k}$-coreset of $C_{i-1,2j-1}$ and $C_{i-1,2j}$ for each $j \in \left[\frac{n}{2^{i-1}}\right]$. By Lemma D.2, it holds that for a fixed $j$, $C_{i,j}$ is an $\frac{\varepsilon}{2k}$-coreset of $C_{i-1,2j-1}$ and $C_{i-1,2j}$ with probability at least $1 - \delta$. By a union bound over $\frac{n}{2^{i-1}}$ possible indices $j$, we have that for a fixed $i$, all $C_{i,j}$ are $\frac{\varepsilon}{2k}$-coresets of $C_{i-1,2j-1}$ and $C_{i-1,2j}$ with probability at least $1 - \frac{n}{2^{i-1}} \cdot \delta$. Thus, $\mathbf{Pr}\left[\mathcal{E}_{i+1}\right] \geq 1 - \frac{n\delta}{2^{i-1}}$, which completes the induction. Hence with $\mathbf{Pr}\left[\cup_{i=0}^k \mathcal{E}_i\right]$, we have that the cost induced by $C_{k,1}$ is a $\left(1 + \frac{\varepsilon}{2k}\right)^k$-approximation to the cost induced by $P$. Since $\left(1 + \frac{\varepsilon}{2k}\right)^k \leq e^{\varepsilon/2} \leq 1 + \varepsilon$, then $C_{k,1}$ is an $\varepsilon$-coreset of $P$ with probability $\mathbf{Pr}\left[\cup_{i=0}^k \mathcal{E}_i\right]$. By a union bound, we have that $\mathbf{Pr}\left[\cup_{i=0}^k \mathcal{E}_i\right] \geq 1 - \sum_{i=0}^k \left(1 - \mathbf{Pr}\left[\mathcal{E}_{i+1}\right]\right) \geq 1 - 2n\delta$. □

## E. Missing Proofs from Section B

**Theorem E.1 (Freedman's inequality)** *(Freedman, 1975)*
*Suppose $Y_0, Y_1, \ldots, Y_n$ is a scalar martingale with difference sequence $X_1, \ldots, X_n$. Specifically, we initiate $Y_0 = 0$ and set $Y_i = Y_{i-1} + X_i$ for all $i \in [n[$ Let $R \geq |X_t|$ for all $t \in [n]$ with high probability. We define the predictable quadratic variation process of the martingale by $w_k := \sum_{t=1}^k \mathbb{E}_{t-1}\left[X_t^2\right]$, for $k \in [n]$. Then for all $\varepsilon \geq 0$ and $\sigma^2 > 0$, and every $k \in [n]$,*

$$\mathbf{Pr}\left[\max_{t \in [k]} |Y_t| > \varepsilon \text{ and } w_k \leq \sigma^2\right] \leq 2\exp\left(-\frac{\varepsilon^2/2}{\sigma^2 + R\varepsilon/3}\right).$$

We first show robustness of our algorithm by justifying correctness of approximation for $L_p$ norms.

**Lemma E.2** ($L_p$ **subspace embedding**) *Suppose $\varepsilon > \frac{1}{n}$, $p \in \{1, 2\}$, and $C > \kappa^p$, where $\kappa$ is an upper bound on the condition number of the stream. Then Algorithm 1 returns a matrix $\mathbf{M}$ such that for all $\mathbf{x} \in \mathbb{R}^d$,*

$$| \|\mathbf{Mx}\|_p - \|\mathbf{Ax}\|_p | \le \varepsilon \|\mathbf{Ax}\|_p,$$

*with high probability.*

**Proof :**  Consider an arbitrary $\mathbf{x} \in \mathbb{R}^d$ and suppose $\varepsilon \in (0, 1/2)$ with $\varepsilon > \frac{1}{n}$. We claim through induction the stronger statement that $|\|\mathbf{M}_j\mathbf{x}\|_p^p - \|\mathbf{A}_j\mathbf{x}\|_p^p| \le \varepsilon\|\mathbf{A}_j\mathbf{x}\|_p^p$ for all times $j \in [n]$ with high probability. Here $\mathbf{M}_j$ is the matrix consisting of the rows of the input matrix $\mathbf{A}$ that have already been sampled at time $j$ and $\mathbf{A}_j = \mathbf{a}_1 \circ \ldots \circ \mathbf{a}_j$. Note that either $\mathbf{a}_1$ is the zero vector or $p_1 = 1$, so that either way, we have $\mathbf{M}_1 = \mathbf{A}_1$ for our base case. We assume the statement holds for all $j \in [n-1]$ and prove it must hold for $j = n$. We implicitly define a martingale $Y_0, Y_1, \ldots, Y_n$ through the difference sequence $X_1, \ldots, X_n$, where for $j \ge 1$, we set $X_j = 0$ if $Y_{j-1} > \varepsilon\|\mathbf{A}_{j-1}\mathbf{x}\|_p^p$ and otherwise if $Y_{j-1} \le \varepsilon\|\mathbf{A}_{j-1}\mathbf{x}\|_p^p$, we set

$$X_j = \begin{cases} \left(\frac{1}{p_j} - 1\right)|\mathbf{a}_j^\top\mathbf{x}|^p & \text{if } \mathbf{a}_j \text{ is sampled in } \mathbf{M} \\ -|\mathbf{a}_j^\top\mathbf{x}|^p & \text{otherwise.} \end{cases} \quad (1)$$

Since $\mathbb{E}[Y_j|Y_1, \ldots, Y_{j-1}] = Y_{j-1}$, then the sequence $Y_0, \ldots, Y_n$ induced by the differences is indeed a valid martingale. Furthermore, by the design of the difference sequence, we have that $Y_j = \|\mathbf{M}_j\mathbf{x}\|_p^p - \|\mathbf{A}_j\mathbf{x}\|_p^p$.

If $p_j = 1$, then $\mathbf{a}_j$ is sampled in $\mathbf{M}_j$, so we have that $X_j = 0$. Otherwise, we have that

$$\begin{aligned} &[\mathbb{E}[X_j^2|Y_1, \ldots, Y_{j-1}] \\ &= p_j\left(\frac{1}{p_j} - 1\right)^2|\mathbf{a}_j^\top\mathbf{x}|^{2p} + (1 - p_j)|\mathbf{a}_j^\top\mathbf{x}|^{2p} \\ &\le \frac{1}{p_j}|\mathbf{a}_j^\top\mathbf{x}|^{2p}. \end{aligned}$$

For $p_j < 1$, then we have $p_j = \alpha\tau_j$ and thus $\mathbb{E}[X_j^2|Y_1, \ldots, Y_{j-1}] \le \frac{1}{\alpha\tau_j}|\mathbf{a}_j^\top\mathbf{x}|^{2p}$. By the definition of $\tau_j$ and the inductive hypothesis that $|\|\mathbf{M}_{j-1}\mathbf{x}\|_p^p - \|\mathbf{A}_{j-1}\mathbf{x}\|_p^p| \le \varepsilon\|\mathbf{A}_{j-1}\mathbf{x}\|_p^p < \frac{1}{2}\|\mathbf{A}_{j-1}\mathbf{x}\|_p^p$, then we have

$$\begin{aligned} \tau_j &\ge \frac{2|\mathbf{a}_j^\top\mathbf{x}|^p}{\|\mathbf{M}_{j-1}\mathbf{x}\|_p^p + |\mathbf{a}_j^\top\mathbf{x}|^p} \\ &\ge \frac{|\mathbf{a}_j^\top\mathbf{x}|^p}{\|\mathbf{A}_{j-1}\mathbf{x}\|_p^p + |\mathbf{a}_j^\top\mathbf{x}|^p} = \frac{|\mathbf{a}_j^\top\mathbf{x}|^p}{\|\mathbf{A}_j\mathbf{x}\|_p^p} \ge \frac{|\mathbf{a}_j^\top\mathbf{x}|^p}{\|\mathbf{Ax}\|_p^p}. \end{aligned}$$

Thus,

$$\begin{aligned} \sum_{j=1}^n \mathbb{E}[X_j^2|Y_1, \ldots, Y_{j-1}] &\le \sum_{j=1}^n \frac{\|\mathbf{Ax}\|_p^p \cdot |\mathbf{a}_j^\top\mathbf{x}|^p}{\alpha} \\ &\le \frac{\|\mathbf{Ax}\|_p^{2p}}{\alpha}. \end{aligned}$$

Moreover, we have that $|X_j| \le \frac{1}{p_j}|\mathbf{a}_j^\top\mathbf{x}|^p$. For $p_j = 1$, we have $\frac{1}{p_j}|\mathbf{a}_j^\top\mathbf{x}|^p \le \|\mathbf{A}_j\mathbf{x}\|_p^p \le \|\mathbf{Ax}\|_p^p$. For $p_j < 1$, we have $p_j = \alpha\tau_j < 1$. Again by the definition of $\tau_j$ and by the inductive hypothesis that $|\|\mathbf{M}_{j-1}\mathbf{x}\|_p^p - \|\mathbf{A}_{j-1}\mathbf{x}\|_p^p| \le \varepsilon\|\mathbf{A}_{j-1}\mathbf{x}\|_p^p < \frac{1}{2}\|\mathbf{A}_{j-1}\mathbf{x}\|_p^p$, we have that

$$\frac{|\langle\mathbf{a}_j, \mathbf{x}\rangle|^p}{2\|\mathbf{A}_j\mathbf{x}\|_p^p} \le \frac{|\langle\mathbf{a}_j, \mathbf{x}\rangle|^p}{\|\mathbf{M}_{j-1}\mathbf{x}\|_p^p + |\langle\mathbf{a}_j, \mathbf{x}\rangle|^p} \le \tau_j.$$

Hence for $\alpha = \frac{Cd}{\varepsilon^2}\log n$, it follows that

$$\begin{aligned} |X_j| &\le \frac{1}{p_j}|\mathbf{a}_j^\top\mathbf{x}|^p \le \frac{2}{\alpha}\|\mathbf{A}_j\mathbf{x}\|_p^p \\ &\le \frac{2\varepsilon^2}{Cd\log n}\|\mathbf{A}_j\mathbf{x}\|_p^p \le \frac{2\varepsilon^2}{Cd\log n}\|\mathbf{Ax}\|_p^p. \end{aligned}$$

We would like to apply Freedman's inequality (Theorem E.1) with $\sigma^2 = \frac{\|\mathbf{Ax}\|_p^{2p}}{\alpha}$ for $\alpha = \mathcal{O}\left(\frac{d}{\varepsilon^2}\log n\right)$ and $R \le \frac{2\varepsilon^2}{d\log n}\|\mathbf{Ax}\|_p^p$, as in (Braverman et al., 2020). However, in the adversarial setting we won't be able bound the probability that $|Y_n|$ exceeds $\varepsilon\|\mathbf{Ax}\|_p^p$ using Freedman's inequality as the latter is a random variable. Thus we instead assume that $\kappa_1, \kappa_2$ are constants so that for $p = 1$, we have $\kappa_1$ and $\kappa_2$ are lower and upper bounds on $\|\mathbf{A}\|_1$ and for $p = 2$, we have that $\kappa_1$ and $\kappa_2$ are lower and upper bounds on the singular values of $\mathbf{A}$. We are now ready to apply Freedman's inequality with $\sigma^2 \le \frac{\kappa_2^{2p}\|\mathbf{x}\|_p^{2p}}{\alpha}$ for $\alpha = \mathcal{O}\left(\frac{d}{\varepsilon^2}\log n\right)$ and $R \le \frac{2\varepsilon^2}{d\log n}\kappa_2^p\|\mathbf{x}\|_p^p$. By Freedman's inequality, we have that

$$\begin{aligned} \mathbf{Pr}\left[|Y_n| > \varepsilon\kappa_1^p\|\mathbf{x}\|_p^p\right] &\le 2\exp\left(-\frac{\kappa_1^{2p}\varepsilon^2\|\mathbf{x}\|_p^{2p}/2}{\sigma^2 + R\kappa_1^p\varepsilon\|\mathbf{x}\|_p^p/3}\right) \\ &\le 2\exp\left(-\frac{3Cd\kappa_1^{2p}\log n/2}{6\kappa_2^{2p} + 2\kappa_1^p\kappa_2^p}\right) \\ &\le \frac{1}{2^d\operatorname{poly}(n)}, \end{aligned}$$

for sufficiently large $C > (\kappa_1/\kappa_2)^p$. Note for $p = 2$, we have the upper bound on the condition number $\kappa \ge \kappa_1/\kappa_2$ so it suffices to set $C = \kappa^2$. Since $\kappa_1^p\|x\|_p^p \le \|\mathbf{Ax}\|_p^p$, then we have

$$\mathbf{Pr}\left[|Y_n| > \varepsilon\|Ax\|_p^p\right] \le \mathbf{Pr}\left[|Y_n| > \varepsilon\kappa_1^p\|\mathbf{x}\|_p^p\right].$$

Thus $|\|\mathbf{Mx}\|_p^p - \|\mathbf{Ax}\|_p^p| \le \varepsilon\|\mathbf{Ax}\|_p^p$ with probability at least $1 - \frac{1}{2^d\operatorname{poly}(n)}$. By a rescaling of $\varepsilon$ since $p \le 2$, we thus have that $|\|\mathbf{Mx}\|_p - \|\mathbf{Ax}\|_p| \le \varepsilon\|\mathbf{Ax}\|_p$ with probability at least $1 - \frac{1}{2^d\operatorname{poly}(n)}$.

We now show that we can union bound over an $\varepsilon$-net. We first define the unit ball $B = \{\mathbf{Ay} \in \mathbb{R}^n \mid \|\mathbf{Ay}\|_p = 1\}$. We also define $\mathcal{N}$ to be a greedily constructed $\varepsilon$-net of $B$.

Since balls of radius $\frac{\varepsilon}{2}$ around each point cannot overlap, but must all fit into a ball of radius $1 + \frac{\varepsilon}{2}$, then it follows that $\mathcal{N}$ has at most $\left(\frac{3}{\varepsilon}\right)^d$ points. Therefore, by a union bound for $\frac{1}{\varepsilon} < n$, we have $| \|\mathbf{My}\|_p - \|\mathbf{Ay}\|_p | \leq \varepsilon \|\mathbf{Ay}\|_p$ for all $\mathbf{Ay} \in \mathcal{N}$, with probability at least $1 - \frac{1}{\text{poly}(n)}$.

We now argue that accuracy on this $\varepsilon$-net implies accuracy everywhere. Indeed, consider any vector $\mathbf{z} \in \mathbb{R}^d$ normalized to $\|\mathbf{Az}\|_p = 1$. We shall inductively define a sequence $\mathbf{Ay}_1, \mathbf{Ay}_2, \ldots$ such that $\left\|\mathbf{Az} - \sum_{j=1}^{i} \mathbf{Ay}_j\right\|_p \leq \varepsilon^i$ and there exists some constant $\gamma_i \leq \varepsilon^{i-1}$ with $\frac{1}{\gamma_i}\mathbf{Ay}_i \in \mathcal{N}$ for all $i$. Define our base point $\mathbf{Ay}_1$ to be the closest point to $\mathbf{Az}$ in the $\varepsilon$-net $\mathcal{N}$. Then since $\mathcal{N}$ is a greedily constructed $\varepsilon$-net, we have that $\|\mathbf{Az} - \mathbf{Ay}_1\|_p \leq \varepsilon$. Given a sequence $\mathbf{Ay}_1, \ldots, \mathbf{Ay}_{i-1}$ such that $\gamma_i := \left\|\mathbf{Az} - \sum_{j=1}^{i-1}\mathbf{Ay}_j\right\|_p \leq \varepsilon^{i-1}$, note that $\frac{1}{\gamma_i}\left\|\mathbf{Az} - \sum_{j=1}^{i-1}\mathbf{Ay}_j\right\|_p = 1$. Thus we inductively define the point $\mathbf{Ay}_i \in \mathcal{N}$ so that $\mathbf{Ay}_i$ is within distance $\varepsilon$ of $\mathbf{Az} - \sum_{j=1}^{i-1}\mathbf{Ay}_j$. Therefore,

$$| \|\mathbf{Mz}\|_p - \|\mathbf{Az}\|_p | \leq \sum_{i=1}^{\infty} | \|\mathbf{My}_i\|_p - \|\mathbf{Ay}_i\|_p |$$
$$\leq \sum_{i=1}^{\infty} \varepsilon^i \|\mathbf{Ay}_i\|_p$$
$$= \mathcal{O}\left(\varepsilon\right) \|\mathbf{Az}\|_p,$$

which completes the induction for time $n$. $\square$

### E.1. Adversarially Robust Spectral Approximation

We observe that Lemma E.2 provides adversarial robustness for free.

**Lemma E.3 (Adversarially robust spectral approximation)** *Algorithm 1 is adversarially robust.*

**Proof :** Let us inspect the proof of Lemma E.2. Observe that since the adversary can observe the past data and the past randomness of Algorithm 1, then the rows $\mathbf{a}_i$ are random variables that depend on the history and the randomness of the algorithm. In other words, $\mathbf{a}_i$ is measurable with respect to the sigma algebra generated by $\mathbf{a}_1, \ldots, \mathbf{a}_{i-1}, B_1, \ldots, B_{i-1}, C_i$ where $B_i$ is the indicator of the event that we sample row $a_i$ in Algorithm 1 and $C_i$ is the random vector generated by the adversary at step $i$ to create row $a_i$.

Denote $\mathfrak{F}_i$ the sigma algebra generated by $a_1, \ldots, a_{i-1}, B_1, \ldots B_{i-1}, C_i$. Then $a_i$ is measurable with respect to $\mathfrak{F}_i$. Let us remind that $Y_j = Y_{j-1} + X_j$ and let us observe that the definition of $X_j$ in Equation (1)

can be rewritten as

$$\left(\left(\frac{1}{p_j} - 1\right)|\mathbf{a}_j^\top \mathbf{x}|^p B_j - |\mathbf{a}_j^\top \mathbf{x}|^p (1 - B_j)\right)\mathbb{I}_{Y_{j-1} < \varepsilon}.$$

It can be easily checked that

$$\mathbb{E}\left[X_j | \mathfrak{F}_j\right]$$
$$= \left(\left(\frac{1}{p_j} - 1\right)|\mathbf{a}_j^\top \mathbf{x}|^p p_j - |\mathbf{a}_j^\top \mathbf{x}|^p (1 - p_j)\right)\mathbb{I}_{Y_{j-1} < \varepsilon}$$
$$= 0.$$

This is because $Y_{j-1}, p_j, a_j$ are measurable with respect to $\mathfrak{F}_j$. This implies that in the adversarial setting sequence $Y_j$ is a martingale with respect to the filtration $\mathfrak{F}_0 \subset \mathfrak{F}_1 \subset \cdots \subset \mathfrak{F}_n$.

The remainder of the proof of Lemma E.2 goes through as is for arbitrary rows $\mathbf{a}_i$'s. Thus, the algorithm is indeed adversarially robust. $\square$

We note the established upper bounds on the sum of the online $L_p$ sensitivities, e.g., Theorem 2.2 in (Cohen et al., 2016), Lemma 2.2 and Lemma 4.7 in (Braverman et al., 2020).

**Lemma E.4 (Bound on Sum of Online $L_p$ Sensitivities)** *(Cohen et al., 2016; Braverman et al., 2020) Let the rows of $\mathbf{A} = \mathbf{a}_1 \circ \ldots \circ \mathbf{a}_n \in \mathbb{R}^{n \times d}$ arrive in a stream with condition number at most $\kappa$ and let $\ell_i$ be the online $L_p$ sensitivity of $\mathbf{a}_i$. Then $\sum_{i=1}^{n} \ell_i = \mathcal{O}\left(d \log n \log \kappa\right)$ for $p = 1$ and $\sum_{i=1}^{n} \ell_i = \mathcal{O}\left(d \log \kappa\right)$ for $p = 2$.*

We note that $\kappa$ is an adversarially chosen parameter, since the rows of the input matrix $\mathbf{A}$ are generated by an adversary. One can mitigate possible adversarial space attacks by tracking $\kappa$ and aborting if $\log \kappa$ exceeds a desired threshold.

**Proof of Lemma B.7:** Algorithm 1 is adversarially robust by Lemma E.3. It remains to analyze the space complexity of Algorithm 1. By Lemma E.3 and a union bound over the $n$ rows in the stream, each row $\mathbf{a}_i$ is sampled with probability at most $4\alpha\tau_i$, where $\tau_i$ is the online leverage score of row $\mathbf{a}_i$. By Lemma E.4, we have $\sum_{i=1}^{n} \tau_i = \mathcal{O}\left(d \log \kappa\right)$ and we also set $\alpha = \mathcal{O}\left(\frac{d\kappa}{\varepsilon^2}\log n\right)$. Let $\gamma > 0$ be a sufficiently large constant such that $\sum_{i=1}^{n} \alpha\tau_i \leq \frac{d^2 \gamma \kappa \log \kappa}{\varepsilon^2}\log n$.

We use a martingale argument to bound the number of rows that are sampled. Consider a martingale $U_0, U_1, \ldots, U_n$ with difference sequence $W_1, \ldots, W_n$, where for $j \geq 1$, we set $W_j = 0$ if $U_{j-1} > \frac{d^2 \gamma \kappa \log \kappa}{\varepsilon^2}\log n$ and otherwise if $U_{j-1} \leq \frac{d^2 \gamma \kappa \log \kappa}{\varepsilon^2}\log n$, we set

$$W_j = \begin{cases} 1 - p_j & \text{if } \mathbf{a}_j \text{ is sampled in } \mathbf{M} \\ -p_j & \text{otherwise.} \end{cases} \quad (2)$$

We have $\mathbb{E}\left[U_j|U_1,\ldots,U_{j-1}\right] = U_{j-1}$, then the sequence $U_0,\ldots,U_n$ induced by the differences is indeed a valid martingale. Note that intuitively, $U_n$ is the difference between the number of sampled rows and $\sum_{j=1}^n p_j$.

Since $\mathbf{a}_j$ is sampled with probability $p_j \in [0,1]$,

$$\mathbb{E}\left[W_j^2|U_1,\ldots,U_{j-1}\right] \leq \sum_{j=1}^n p_j \leq \sum_{j=1}^n \alpha \tau_j.$$

Moreover, we have $\mathbb{E}\left[|W_j|\,|\,U_1,\ldots,U_{j-1}\right] \leq 1$. Thus by Freedman's inequality (Theorem E.1) with $\sigma^2 = \sum_{j=1}^n \alpha\tau_j \leq \frac{d^2\gamma\kappa\log\kappa}{\varepsilon^2}\log n$ and $R \leq 1$,

$$\mathbf{Pr}\left[|U_n| > \frac{d^2\gamma\kappa\log\kappa}{\varepsilon^2}\log n\right]$$

$$\leq 2\exp\left(-\frac{d^4\gamma^2\kappa^2\log^2\kappa\log^2 n/(2\varepsilon^4)}{\sigma^2 + Rd^2\gamma\kappa\log\kappa\log n/(3\varepsilon^2)}\right)$$

$$\leq \frac{1}{\mathrm{poly}(n)}.$$

Hence we have that with high probability, the number of rows sampled is $\mathcal{O}\left(\frac{1}{\varepsilon^2}d^2\kappa\log\kappa\log n\right)$. $\square$

We remark that the space bounds for Lemma B.7 could similarly be shown (with constant probability of success) using Markov's inequality though analysis Freedman's inequality provides much higher guarantees in terms of probability of success.

On the other hand, it is not clear how to execute a similar strategy using the Matrix Freedman's Inequality rather than using Freedman's inequality. This is because to obtain the desired spectral bound, we must define a martingale at time $j$ in terms of both the matrix $\mathbf{A}_j$ and whether the rows $\mathbf{a}_1,\ldots,\mathbf{a}_{j-1}$ were previously sampled. However, since $\mathbf{A}_j$ is itself a function of whether $\mathbf{a}_1,\ldots\mathbf{a}_{j-1}$ were previously sampled, the resulting sequence is not a valid martingale.

We first require the following bound on the sum of the online ridge leverage scores, e.g., Theorem 2.12 from (Braverman et al., 2020), which results from considering Lemma 2.11 in (Braverman et al., 2020) at $\mathcal{O}(\log n)$ different scales.

**Lemma E.5** *(**Bound on Sum of Online Ridge Leverage Scores**) (Braverman et al., 2020) Let the rows of $\mathbf{A} = \mathbf{a}_1 \circ \ldots \circ \mathbf{a}_n \in \mathbb{R}^{n\times d}$ arrive in a stream with condition number at most $\kappa$, let $\lambda_i = \frac{\|\mathbf{A}_i - (\mathbf{A}_i)_{(k)}\|_F^2}{k}$, where $\mathbf{A}_i = \mathbf{a}_1 \circ \ldots \circ \mathbf{a}_i$ and $(\mathbf{A}_i)_{(k)}$ is the best rank $k$ approximation to $\mathbf{A}_i$. Let $\ell_i$ be the online ridge leverage score of $\mathbf{a}_i$ with regularization $\lambda_i$. Then $\sum_{i=1}^n \ell_i = \mathcal{O}(k\log n\log\kappa)$.*

From Lemma E.5 and a similar argument to Lemma E.2, we also obtain adversarially robust projection-cost preservation and therefore low-rank approximation. Namely, (Cohen

et al., 2017; Braverman et al., 2020) showed that projection-cost preservation essentially reduces to sampling a weighted submatrix $\mathbf{M}$ of $\mathbf{A}$ such that $\|\mathbf{Mx}\|_2^2 + \lambda\|x\|_2^2 \in (1 \pm \varepsilon)(\|\mathbf{Ax}\|_2^2 + \lambda\|x\|_2^2)$ for a ridge parameter $\lambda$. Since the online ridge leverage score of each row $\mathbf{a}_i$ can be rewritten as $\max_{\mathbf{x}\in\mathbb{R}^d}\frac{\langle\mathbf{a}_i,x\rangle^2+\lambda\|\mathbf{x}\|_2^2}{\|\mathbf{A}_i\mathbf{x}\|_2^2+\lambda\|x\|_2^2}$, then the same concentration argument of Lemma E.2 gives Lemma B.8.

### E.2. Adversarially Robust Linear Regression

We first give the formal definition of linear regression:

**Problem E.6 (Linear Regression)** *Given a matrix $\mathbf{A} \in \mathbb{R}^{n\times d}$, a vector $\mathbf{b} \in \mathbb{R}^n$ and an approximation parameter $\varepsilon > 0$, the goal is to output a vector $\mathbf{y}$ such that $\|\mathbf{Ay} - \mathbf{b}\|_2 \leq (1+\varepsilon)\min_{\mathbf{x}\in\mathbb{R}^n}\|\mathbf{Ax} - \mathbf{b}\|_2$.*

**Lemma E.7 (Adversarially Robust Linear Regression)** *Given $\varepsilon > 0$ and a matrix $\mathbf{A} \in \mathbb{R}^{n\times d}$ whose rows $\mathbf{a}_1,\ldots,\mathbf{a}_n$ arrive sequentially in a stream with condition number at most $\kappa$, there exists an adversarially robust streaming algorithm that outputs a $(1+\varepsilon)$ approximation to linear regression and uses $\mathcal{O}\left(\frac{d^3}{\varepsilon^2}\log^2 n\log\kappa\right)$ bits of space, with high probability.*

**Proof :** Suppose each row of $\mathbf{A}$ arrives sequentially, along with the corresponding entry in $\mathbf{b}$. Let $\mathbf{B} = \mathbf{A} \circ \mathbf{b}$ so that the effectively, the rows of $\mathbf{B}$ arrive sequentially. Note that if $\mathbf{M}$ is a spectral approximation to $\mathbf{B}$, then we have

$$(1-\varepsilon)\|\mathbf{Bv}\|_2 \leq \|\mathbf{Mv}\|_2 \leq (1+\varepsilon)\|\mathbf{Bv}\|_2$$

for all vectors $\mathbf{v} \in \mathbb{R}^{d+1}$. In particular, let $\mathbf{w} \in \mathbb{R}^{d+1}$ be the vector that minimizes $\|\mathbf{Mv}\|_2$ subject to the constraint that the last coordinate of $\mathbf{w}$ is 1, and let $\mathbf{w} = \begin{bmatrix}\mathbf{y}\\1\end{bmatrix}$. Then we have

$$\|\mathbf{Ay} - \mathbf{b}\|_2 = \|\mathbf{Bw}\|_2 \leq \frac{1}{1-\varepsilon}\|\mathbf{Mw}\|_2.$$

Let $\mathbf{z}$ be the vector that minimizes $\|\mathbf{Ax} - \mathbf{b}\|_2$ and let $\mathbf{u} = \begin{bmatrix}\mathbf{z}\\1\end{bmatrix}$. Then we have

$$\|\mathbf{Az} - \mathbf{b}\|_2 = \|\mathbf{Bu}\|_2 \geq \frac{1}{1+\varepsilon}\|\mathbf{Mu}\|_2 \geq \frac{1}{1+\varepsilon}\|\mathbf{Mw}\|_2,$$

where the last inequality follows from the minimality of $\mathbf{w}$. Thus we have that $\|\mathbf{Ay} - \mathbf{b}\|_2 \leq (1+\mathcal{O}(\varepsilon))\|\mathbf{Az} - \mathbf{b}\|_2$. $\square$

## F. Missing Proofs from Section C

**Other Related Works** Note that there is an alternate streaming algorithm for graph sparsification given in (Goel

et al., 2010) which has the same guarantees but is computationally faster. However, we choose to analyze the algorithm of (Ahn & Guha, 2009) since its core argument is sampling based. Nevertheless, it is possible that the algorithm from (Goel et al., 2010) is also adversarially robust. Lastly, we recall that our model is the streaming model where edges arrive one at a time. There is also related work in the dynamic streaming model (see (Kapralov et al., 2019) and references therein) where previously shown edges can be deleted but this is not the scope of our work.

The notion of the connectivity of an edge is needed to in the algorithm of (Ahn & Guha, 2009).

**Definition F.1 (Connectivity (Benczúr & Karger, 1996))**
*A graph is $k$-strong connected iff every cut in the graph has value at least $k$. A $k$-strong connected component is a maximal node-induced subgraph which is $k$-strong connected. The connectivity of an edge $e$ is the maximum $k$ such that there exists a $k$-strong connected component that contains $e$.*

---

**Algorithm 2** Graph sparsification algorithm from (Ahn & Guha, 2009).

---

**Input:** A stream of edges $e_1, \cdots, e_m$ and an accuracy parameter $\varepsilon > 0$
**Output:** Sparified graph $H$
1: $H \leftarrow \emptyset$
2: $\rho \leftarrow C(\log n + \log m)/\varepsilon^2$ for sufficiently large constant $C > 0$
3: **for** each new edge $e$ **do**
4:     compute the connectivity $c_e$ of $e$ in $H$
5:     $p_e = \min(\rho/c_e, 1)$     ▷Importance of edge $e$, see Definition F.1
6:     Add $e$ to $H$ with probability $p_e$ and weight $1/p_e$ times its original weight
7: **return** $H$

---

We begin by providing a brief overview of our proof. The first step is to show that for a cut in $G$ of value $c$, the same cut in the sparsified graph $H$ has value that concentrates around $c$. Note that in (Ahn & Guha, 2009), the concentration inequality they obtain *depends* roughly on $\exp(-c)$. In other words, they get a stronger concentration for larger cuts in the original graph. However, their concentration inequality is not valid in our setting since the value $c$ is *random*. Therefore, we employ a different concentration inequality, namely Freedman's inequality (Theorem E.1) in conjunction with an assumption about the sizes of cuts in the graph to obtain concentration for a fixed cut. The second step is to use a standard worst-case union bound strategy to bound the total number of cuts with a particular size in the original graph. This uses the standard fact that the number of cuts in a graph that is at most $\alpha$ times the minimum cut

is at most $n^{2\alpha}$. Then the final result for the property (1) in Problem C.1 follows by combining the union bound with the previously mentioned concentration inequality. The bound for the total number of edges (condition (2) in Problem C.1) is a "worst case" calculation in (Ahn & Guha, 2009) so it automatically ports over to our setting. Note that we assume $\kappa_1$ and $\kappa_2$ to be deterministic lower and upper bounds on the size of any cut in $G$ and define $\kappa$ to be their ratio.

**Theorem C.2** *Given a weighted graph $G = (V, E)$ with $|V| = n$ whose edges $e_1, \ldots, e_m$ arrive sequentially in a stream, there exists an adversarially robust streaming algorithm that outputs a $1 \pm \varepsilon$ cut sparsifier with $\mathcal{O}\left(\frac{\kappa^2 n \log n}{\varepsilon^2}\right)$ edges with probability $1 - 1/\operatorname{poly}(n)$.*

**Proof :** We claim through induction the stronger statement that the value $C_H$ of any cut in $H$ is a $(1 + \varepsilon)$-approximation of the value $C_G$ of the corresponding cut in $G$ for all times $j \in [m]$ with high probability. Consider a fixed set $S \subseteq V$ and the corresponding cut $C = (S, V \setminus S)$. Let $e_1, \ldots, e_m$ be the edges of the stream in the order that they arrive. We emphasize that $e_1, \ldots, e_m$ are possibly random variables given by the adversary rather than fixed edges. For each $j \in [m]$, let $G_j$ be the graph consisting of the edges $e_1, \ldots, e_j$ and let $H_j$ be the corresponding sampled weighted subgraph. We abuse notation and define $p_j := p_{e_j}$ to denote the probability of sampling the edge $e_j$ that arrives at time $j$. We use $C_G^{(j)}$ and $C_H^{(j)}$ to denote the value of the cut at time $j$ in graphs $G$ and $H$, respectively. Note that $p_1 = 1$, so we have $H_1 = G_1$ for our base case.

We assume the statement holds for all $j \in [m - 1]$ and prove it must hold for $j = m$. We define a martingale $Y_0, Y_1, \ldots, Y_m$ through its difference sequence $X_1, \ldots, X_m$, where for $j \geq 1$, we set $X_j = 0$ if $C_H^{(j-1)} \notin (1 \pm \varepsilon)C_G^{(j-1)}$. Otherwise if $(1 - \varepsilon)C_G^{(j-1)} \leq C_H^{(j-1)} \leq (1+\varepsilon)C_G^{(j-1)}$, then we set $X_j$ equal to 0 if $e_j$ does not cross the cut $C$, $\left(\frac{1}{p_j} - 1\right)$ if $e_j$ crosses the cut and is sampled in $H$, and $-1$ if $e_j$ crosses the cut and is not sampled in $H$.

Because $\mathbb{E}[Y_j | Y_1, \ldots, Y_{j-1}] = Y_{j-1}$, then we have that the sequence $Y_0, \ldots, Y_n$ is indeed a valid martingale and that $Y_j = C_H^{(j)} - C_G^{(j)}$. (We abuse notation and use $Y_1, \ldots, Y_i$ to indicate the similar filtration to the one in Lemma Lemma E.3).

If $p_j = 1$, then $e_j$ is sampled in $H_j$, so we have that $X_j = 0$. Otherwise,

$$\mathbb{E}\left[X_j^2 | Y_1, \ldots, Y_{j-1}\right] = p_j \left(\frac{1}{p_j} - 1\right)^2 + (1 - p_j) \leq \frac{1}{p_j}.$$

For $p_j < 1$, then we have $p_j = \rho/c_{e_j}$ and thus $\mathbb{E}\left[X_j^2 | Y_1, \ldots, Y_{j-1}\right] \leq \frac{c_{e_j}}{\rho}$. Thus,

$\sum_{j=1}^n \mathbb{E}\left[X_j^2 | Y_1, \ldots, Y_{j-1}\right] \leq \sum_{j:e_j \in C} \frac{c_{e_j}}{\rho}$. Recall that $c_{e_j}$ is the connectivity of $e_j$ in $H$ rather than $G$. However, by the definition of $c_{e_j}$ and the inductive hypothesis that $H_{j-1}$ is a $(1 + \varepsilon)$ cut sparsifier of $G_{j-1}$, then we have that for $\varepsilon < \frac{1}{2}$, the connectivity of $c_{e_j}$ in $H$ is within a factor of two of the connectivity of $c_{e_j}$ in $G$. By definition of connectivity, we have that the connectivity of $c_{e_j}$ at time $j$ in $G$ is at most $C_G^{(j)} \leq C_G^{(m)}$ if $e_j$ crosses the cut $C$. Hence,

$$\sum_{j=1}^m \mathbb{E}\left[X_j^2 | Y_1, \ldots, Y_{j-1}\right] \leq \sum_{j:e_j \in C} \frac{C_G^{(j)}}{\rho} \leq \frac{2(C_G^{(m)})^2}{\rho}.$$

By similar reasoning, we have $|X_j| \leq \frac{1}{p_j} \leq \frac{c_{e_j}}{\rho} \leq \frac{2(C_G^{(m)})}{\rho}$. Now we would like to apply Freedman's inequality (Theorem E.1) with $\sigma^2 = \frac{2(C_G^{(m)})^2}{\rho}$ and $R \leq \frac{2(C_G^{(m)})}{\rho}$ for $\rho = C(\log n + \log m)/\varepsilon^2$. However, we cannot bound the probability that $|Y_n|$ exceeds $\varepsilon C_G^{(m)}$, as the latter is a random variable. Thus we instead assume that $\kappa_1$ and $\kappa_2$ are lower and upper bounds on $C_G^{(m)}$. By Freedman's inequality,

$$\mathbf{Pr}\left[|Y_n| > \varepsilon \kappa_1\right] \leq 2\exp\left(-\frac{\kappa_1^2 \varepsilon^2/2}{\sigma^2 + R\kappa_1 \varepsilon/3}\right)$$

$$\leq 2\exp\left(-\frac{3C\kappa_1^2 \log n/2}{6\kappa_2^2 + 2\kappa_1 \kappa_2}\right)$$

$$\leq n^{-O(C/\kappa^2)},$$

where we define $\kappa := \kappa_2/\kappa_1$. Since $\kappa_1 \leq C_G^{(m)}$, then we have

$$\mathbf{Pr}\left[|Y_n| > \varepsilon C_G^{(m)}\right] \leq \mathbf{Pr}\left[|Y_n| > \varepsilon \kappa_1\right].$$

Thus $|C_H^{(m)} - C_G^{(m)}| \leq \varepsilon C_G^{(m)}$ with probability at least $1 - n^{-O(C/\kappa^2)}$.

We now union bound over all cuts $C$. Based on our assumption that every cut in $G$ has value at least $\kappa_1$, it follows that for any $\alpha \geq 1$, the number of cuts in $G$ of size $\alpha \kappa_1$ is at most $n^{2\alpha}$ (Benczúr & Karger, 1996; Ahn & Guha, 2009). Note that we are using a *deterministic* upper bound on the number of cuts that holds for any graph. Due to our assumption, on the size of cuts, we know that $\alpha$ ranges from $1 \leq \alpha \leq \kappa_2/\kappa_1 = \kappa$. Then using our concentration result derived above, it follows by a union bound that the probability that there exists some $C$ such that $|C_H^{(m)} - C_G^{(m)}| \leq \varepsilon C_G^{(m)}$ is at most

$$\int_1^{\kappa_2/\kappa_1} n^{2\alpha} \cdot n^{-O(C/\kappa^2)} \, d\alpha \leq \frac{n^{2\kappa}}{2\log(\kappa)} \cdot n^{-O(C/\kappa^2)}$$

$$\leq \frac{1}{\text{poly}(n)}$$

where the last inequality follows by setting $C = c'\kappa^2$ for some large enough constant $c' > 1$. This verifies part (1) of Problem C.1.

We now need to check the number of edges in $H$. For this, we note that the proof of Theorem 3.2 in (Ahn & Guha, 2009) carries over to our setting since the proof there only relies on the fact that if an edge has strong connectivity at most $z$ in $G$, its weight in $H$ is at most $z/\rho$ in $H$ which is true for us as well. The extra $\kappa^2$ factor in the number of edges comes from our setting of the parameter $C$ in $\rho$. □