# OpenReview forum: "Adversarial Robustness of Streaming Algorithms through Importance Sampling"
_ICML.cc/2021/Workshop/AML — ICML 2021 Workshop AML Oral_

### Official Review · Reviewer_VpG8 · 2021-06-20
**Interesting work**

**Rating:** Accept
**Confidence:** 4

**Review:**

As far as I understand, this paper proves that several meta-approaches used in streaming algorithms (e.g., merge and reduce, row sampling) can prevent adversaries from affecting the independence of importance sampling on $u_t$. The conclusions are well-generalized for many machine learning and numerical algorithms, which is worthy of more attention from the community.

---

### Decision · Program_Chairs · 2021-06-21

**Decision:**

Accept (Oral)

**Comment:**

The reviewer believed that this paper is worthy of more attention from the community. Overall, this is a good paper with solid results.